# SELF-IMPROVING MODEL STEERING

## ABSTRACT

Model steering represents a powerful technique that dynamically aligns large language models (LLMs) with human preferences during inference. However, conventional model-steering methods rely heavily on externally annotated data, not only limiting their adaptability to varying contexts but also tethering their effectiveness to annotation quality. In this paper, we present SIMS, the first self-improving model-steering framework that operates without relying on external supervision. At its core, SIMS autonomously generates and refines contrastive samples through iterative self-improvement cycles, enabling adaptive, context-specific steering. Additionally, SIMS employs novel strategies, including prompt ranking and contrast sampling, to further enhance steering efficacy. Extensive evaluation across diverse LLMs and benchmarks demonstrates that SIMS substantially outperforms existing methods in steering effectiveness and adaptability, highlighting self-improving model steering as a promising direction for future research on inference-time LLM alignment. The code for replicating SIMS is available at `https://anonymous.4open.science/r/SIMS/`

## 1 INTRODUCTION

Model steering (Panickssery et al., 2023; Li et al., 2023; Qiu et al., 2024) represents a compelling alternative to pre- and post-training alignment methods for large language models (LLMs) (Ouyang et al., 2022; Lee et al., 2024b). By modifying latent activations with pre-computed steering vectors on the fly, it enables alignment without expensive retraining. A variety of approaches have been proposed to compute the steering vectors, ranging from linear transformations and projections (Panickssery et al., 2023; Li et al., 2023)

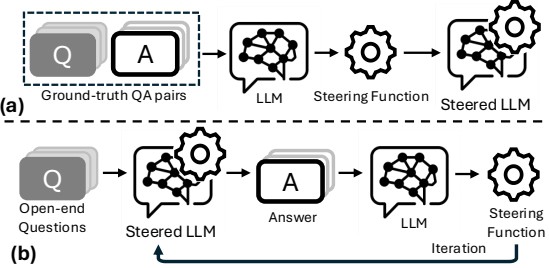

Figure 1: Comparison of (a) conventional and (b) self-improving model steering.

to subspace learning and optimization techniques (Qiu et al., 2024; Pham & Nguyen, 2024; Zhang et al., 2024; Cao et al., 2024). However, as illustrated in Figure 1 (a), most existing methods rely heavily on labeled alignment datasets to optimize steering vectors. This dependence assumes complete prior knowledge of what constitutes good versus bad examples through pre-existing datasets, an assumption with two major limitations. First, it demands access to diverse data sources, including outputs from different LLMs with varying architectures and sizes, or extensive human annotations. Second, it requires high-quality labels that accurately capture response alignment with specific objectives. Together, these requirements significantly limit the practical applicability of model steering.

In this paper, we explore whether we can derive high-quality steering vectors using only the data from the LLM itself. We present SIMS,[1] a model-steering framework that enables model alignment through iterative refinement of the model's own responses. As illustrated in Figure 1 (b), SIMS distinguishes itself from conventional methods through two fundamental innovations. *i*) Self-play steering – SIMS eliminates dependency on external responses and their corresponding labels by leveraging self-generated samples to derive steering directions. This paradigm shift enhances adaptability to varying contexts and data distributions. *ii*) Iterative self-improvement – through cycles of evaluation and

---

[1]SIMS: Self-Improving Model Steering.

regeneration, SIMS continuously refines steering directions to more effectively differentiate desirable and undesirable behaviors, leading to consistent performance gains across iterations. Additionally, we introduce two variants to further enhance steering efficacy: *i*) prompt ranking (SIMS-PR), which leverages the model's own judgment to generate preference signals, eliminating the need for external oracles and enabling fully autonomous self-improvement; and *ii*) contrast sampling (SIMS-CS), which maintains a response bank to select the most informative question-response pairs across iterations, thereby improving sampling efficiency.

Through extensive evaluation across diverse LLMs and benchmarks, we show that SIMS effectively steers LLMs towards desirable behaviors, outperforming or matching existing model-steering methods that rely on externally annotated data. For instance, SIMS improves the length-controlled WinRate of `llama3-8b` on Alpaca-Eval (Dubois et al., 2025) from 2.86 to 11.89 in just one iteration. Our ablation study further reveals that SIMS steadily enhances steering effectiveness across iterations, while SIMS-PR and SIMS-CS substantially improve steering efficiency. For example, SIMS increases the Arena-Hard (Li et al., 2024) score sharply from 15.3 to 33.4 from the first iteration to the third iteration. The findings highlight self-improving model steering as a promising direction for future research on inference-time preference optimization.

Our contributions can be summarized as follows.

- We introduce SIMS, a novel self-improving model-steering framework that iteratively refines steering directions through self-improvement cycles, enabling adaptive, context-specific steering.
- We further implement two variants of SIMS, namely SIMS-PR and SIMS-CS. SIMS-PR leverages the model's own judgment to generate preference signals, while SIMS-CS selects informative samples for refining steering directions.
- We conduct an extensive evaluation to validate that SIMS effectively guides LLMs towards desirable behaviors, consistently outperforming or matching existing methods that require externally annotated data.

## 2 RELATED WORK

**Model Steering.** Unlike pre/post-training alignment (Ouyang et al., 2022; Lee et al., 2024b), model steering modifies latent activations at inference time (Turner et al., 2023; Liu et al., 2023; Zou et al., 2023; Wu et al., 2024c; Chalnev et al., 2024; Lee et al., 2024a; He et al., 2024; Fang et al., 2024; Rodriguez et al., 2024; Wang et al., 2024; Liu et al., 2024a; Cao et al., 2024). Methods differ by how steering vectors are obtained. Linear approaches (Turner et al., 2023; Panickssery et al., 2023; Li et al., 2023) derive vectors from activations: ActADD (Turner et al., 2023) uses activation differences elicited by opposing prompts (e.g., truthful versus deceptive), and CAA (Panickssery et al., 2023) averages differences between paired positive/negative prompts. Nonlinear interventions (Qiu et al., 2024; Zhang et al., 2024; Pham & Nguyen, 2024) act in learned subspaces; e.g., HPR (Pham & Nguyen, 2024) learns global separating hyperplanes and rotations to reflect and rotate activations toward desirable behavior. However, most methods rely on externally annotated data (e.g., question–answer pairs), limiting adaptability and tying effectiveness to annotation quality.

**Preference optimization.** Reinforcement learning from human feedback (RLHF) has emerged as apro, prominent approach for learning human preferences (Ouyang et al., 2022; Lee et al., 2024b). RLHF first trains a reward model on preference data using established frameworks (e.g., the Bradley-Terry model (Huang et al., 2004)), and applies RL algorithms (e.g., PPO (Schulman et al., 2017)) to optimize LLMs with respect to the reward model. Recent work (Rafailov et al., 2023; Zhao et al., 2023) shows the feasibility of bypassing the explicit reward modeling and directly solving the underlying RL problem. Further, SRSO (Liu et al., 2024b) unifies the losses of DPO (Rafailov et al., 2023) and SLiC (Zhao et al., 2023), offering an improved estimate of the optimal policy. This work extends previous research on preference optimization into challenging scenarios where externally annotated data is unavailable or impractical to obtain, addressing a critical gap in current work.

**LLM Self-Improvement.** Self-improvement, in which models generate, judge, and refine their own outputs, can enhance alignment, instruction following, and preference modeling while reducing annotation effort and exposure to harmful content (Chen et al., 2025; Dong et al., 2024b; Song et al., 2024; Subramaniam et al., 2025; Choi et al., 2024; Wu et al., 2024a; Peng et al., 2024; Wan et al., 2025). Approaches include synthetic preference generation (Dong et al., 2024b; Lee et al.,

2024b), tree-search refinement (Cheng et al., 2024; Light et al., 2023), Nash-equilibrium-based optimization (Wu et al., 2024b), execution-guided verification (Dong et al., 2024a), and iterative self-evolved reward modeling (Huang et al., 2024), differing mainly in feedback mechanism and granularity (internal judgment, strategic refinement, external execution validation). To our best knowledge, this work represents the first exploration of this paradigm for model steering.

# 3 PRELIMINARIES

## 3.1 MODEL STEERING

Let $\mathcal{M}$ denote an $L$-layer, Transformer-based LLM and $x$ be a tokenized prompt. The embedding matrix $W_E$ maps tokens to the initial hidden state $h_0 = W_E(x)$. For each layer $l \in [L]$, we apply multi-head attention (MHA) followed by a position-wise feed-forward network (FFN), each with a residual connection:[2]

$$h'_l = h_{l-1} + \text{MHA}_l(h_{l-1}), \quad h_l = h'_l + \text{FFN}_l(h'_l). \tag{1}$$

The model's logits are obtained via $\mathcal{M}(x) = W_U(h_L)$, where $W_U$ is the un-embedding matrix.

During inference, we inject steering functions $f_l$ and $f'_l$ into the residual stream:

$$\tilde{h}'_l = \tilde{h}_{l-1} + \text{MHA}_l(f_l(\tilde{h}_{l-1})), \quad \tilde{h}_l = \tilde{h}'_l + \text{FFN}_l(f'_l(\tilde{h}'_l)), \tag{2}$$

where $f_l$ (respectively $f'_l$) operates immediately before the attention (respectively FFN) while the residual addition preserves the original signal. The steered model then produces $\tilde{\mathcal{M}}(x) = W_U(\tilde{h}_L)$.

Given a dataset $\mathcal{D} = \{(x_i, y_i^+, y_i^-)\}_{i=1}^N$, where $y_i^+$ (desired) and $y_i^-$ (undesired) exhibit opposite attributes, we form positive and negative samples $(x_i, y_i^+)$ and $(x_i, y_i^-)$, respectively. Passing these examples through $\mathcal{M}$ yields paired hidden activation sets:

$$\mathcal{H}_l^+ = \{(h_{l,i}^+, h'^+_{l,i})\}_i, \qquad \mathcal{H}_l^- = \{(h_{l,i}^-, h'^-_{l,i})\}_i. \tag{3}$$

Existing model-steering methods learn $f_l$ and $f'_l$ by exploiting the discrepancy between $\mathcal{H}_l^+$ and $\mathcal{H}_l^-$ using contrastive or other representation-learning objectives (details in §2). We refer to these methods as *steering-function learners* in the following.

## 3.2 SELF-IMPROVEMENT LEARNING

We formalize the self-improvement optimization as follow. Given an LLM $\mathcal{M}$, we prompt $\mathcal{M}$ with input $x$ and obtain two responses $y$ and $y'$.

The self-improvement learning aims to optimize the alignment of $\mathcal{M}$ to human preferences. This process is typically done by reinforcement learning, which $\mathcal{M}$ represents the initial policy $\pi_0$. A preference oracle $\mathcal{O}$, obtained from human feedback, is introduced in the learning process. Given the input $x$ and two responses $y$ and $y'$, The oracle $\mathcal{O}$ will provide preference feedback $o(y \succ y'|x) \in \{0, 1\}$ indicating whether $y$ is preferred over $y'$. We denote $\mathbb{P}(y \succ y'|x) = \mathbb{E}[o(y \succ y'|x)]$ as the probability of $y$ 'winning the duel' over $y'$. In addition, we define the winning probability of $y$ against a distribution of responses from policy $\pi$ as

$$\mathbb{P}(y \succ \pi|x) = \mathbb{E}_{y' \sim \pi(\cdot|x)}[\mathbb{P}(y \succ y'|x)]. \tag{4}$$

The self-improvement learning takes an iterative process to update the policy $\pi_t$, where $t$ denotes the iteration number. For every iteration $t$, $\pi_t$ is optimized based on the objective function as:

$$\pi_{t+1} = \arg\max_\pi \mathbb{E}_{x \sim \mathcal{D}}\left(\mathbb{E}_{y \sim \pi(\cdot|x)}\mathbb{P}(y \succ \pi_t|x)\right). \tag{5}$$

However, the above equation is hard to optimized directly through gradient. Reference probability $\mathbb{P}(y \succ \cdot)$ is typically non-smooth and lead to high-variance. To overcome this shortcomings, many works adpot KL-regularized, max-entropy RL objective as follows

$$\pi_{t+1} = \arg\min_\pi \mathbb{E}_{y \sim \pi_t(\cdot|x)}\left[\left(\log\frac{\pi(y|x)}{\pi_t(y|x)} - \left(\eta\,\mathbb{P}(y \succ \pi_t|x) - \log Z_{\pi_t}(x)\right)\right)^2\right], \tag{6}$$

---

[2]Layer normalization and projection matrices are omitted for clarity.

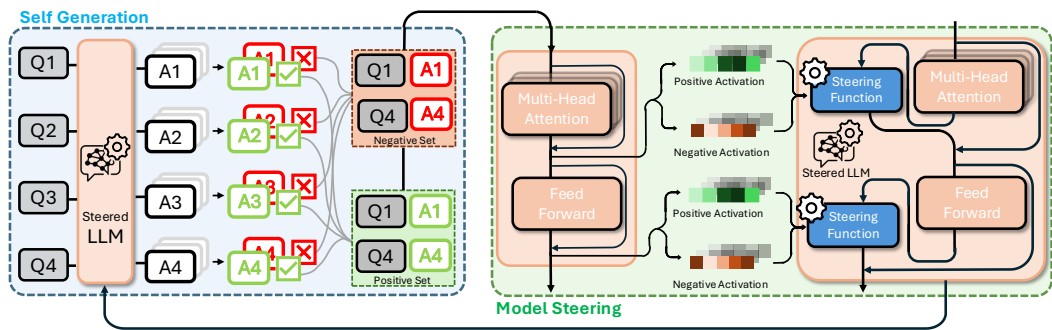

Figure 2: Overview of SIMS. With $N = 4$ questions (prompts) drawn from a prompt distribution $\mathcal{D}_{\text{prompt}}$. We generate the $K = 3$ responses from model inference. We filter the responses into a positive set and a negative set. Running these samples through the LLM, we collect the positive and negative activation sets. These sets are used to update the steering functions by the steering-function learner $\mathcal{A}$. We combine the updated steering functions with the base model to form the refined policy for the next iteration.

where $\log Z_{\pi_t}(x)$ denotes the normalization term. SIMS extends the self-improvement paradigm to model steering, enabling LLMs to introspectively refine internal activations through iterative cycles of self-assessment and enhancement.

## 4 METHOD

Next, we present SIMS, the first self-improving model-steering framework, with its overview illustrated in Figure 2.

### 4.1 SELF-IMPROVING MODEL STEERING

At its core, SIMS autonomously generates and refines contrastive samples through iterative self-improvement cycles, enabling learning the steering function from LLMs' own behaviors without external supervision.

At each iteration $t$, the current steering policy $\pi_{t-1}$ processes a mini-batch of $N$ prompts sampled from the question distribution $\mathcal{D}_q$. For each prompt, the policy produces $K$ candidate responses. A preference oracle $\mathcal{O}$, which could be an existing reward model or even $\pi_{t-1}$ itself acting as its own evaluator, is queried to yield an ordering over the $K$ responses. These preference judgments define *positive* ($\mathcal{D}_t^+$) and *negative* ($\mathcal{D}_t^-$) sample buffers that pair each prompt with its preferred or disfavored outputs, respectively, creating contrastive training signals.

The language model $\mathcal{M}$ is then executed on both positive samples $(x_i, y_i^+)$ from $\mathcal{D}_t^+$ and the negative samples $(x_i, y_i^-)$ from $\mathcal{D}_t^-$. We collect layer-wise activations to construct two activation sets, $\mathcal{H}_l^+$ and $\mathcal{H}_l^-$, as defined in Eq. 3. We leverage an existing steering-function learner $\mathcal{A}$ (e.g., HPR Pham & Nguyen (2024)) to update the steering functions $\{f_l, f_l'\}_{l=1}^L$, which linearly or non-linearly shift model activations toward preferred behaviors while repelling undesirable ones. By composing the updated steering functions with the base model $\mathcal{M}$, we derive the refined policy $\pi_t$ for the next iteration.

The above process is iteratively repeated to progressively refine the steering functions. Because SIMS bootstraps its training signal entirely from its own generated outputs, it decouples model steering from externally annotated data and can be extended through an arbitrary number of iterations $T$. Under mild assumptions about oracle accuracy, the policy sequence $\{\pi_t\}_{t=0}^T$ constitutes monotonic improvement in expected preference reward. Crucially, each update operates only on sub-token activations rather than modifying full model weights, thereby maintaining computational efficiency compared to full-scale fine-tuning.

The complete algorithm is sketched in Algorithm 1.

---

**Algorithm 1:** SELF-IMPROVING MODEL STEERING (SIMS)

---

**Input:** Language model $\mathcal{M}$ with $L$ layers; preference oracle $o$; steering-rule learner $\mathcal{A}$; prompt distribution $D_{\text{prompt}}$; iterations $T$; prompts per iteration $N$; responses per prompt $K$

**Output:** Final steered policy $\pi_T$

1  Initialize steering transforms $\{f_l^{(0)}\}_{l=1}^L$ and $\{f_l'^{(0)}\}_{l=1}^L$;

2  Define initial policy $\pi_0 = (\mathcal{M}, \{f_l^{(0)}\}_{l=1}^L, \{f_l'^{(0)}\}_{l=1}^L)$;

3  **for** $t = 1$ **to** $T$ **do**

4     Sample prompts $\{\mathbf{x}_n\}_{n=1}^N \sim D_{\text{prompt}}$;

5     **for** $n = 1$ **to** $N$ **do**

6        Generate $K$ candidate responses $\{\mathbf{y}_{n,k}\}_{k=1}^K \sim \pi_{t-1}(\cdot \mid \mathbf{x}_n)$;

7        Query oracle $o$ for pairwise preferences $\mathbb{P}_o(\mathbf{y}_{n,k} \succ \mathbf{y}_{n,k'})$ ;        `// All k<k'`

8     Construct datasets
        $\mathcal{D}_t^+ = \{(\mathbf{x}_n, \mathbf{y}_{n,k}) \mid \mathbb{P}(y_{n,k} \succ \boldsymbol{\pi}_t \mid x_n)\}, \quad \mathcal{D}_t^- = \{(\mathbf{x}_n, \mathbf{y}_{n,k}) \mid \mathbb{P}(y_{n,k} \prec \boldsymbol{\pi}_t \mid x_n)\}$;

9     Collect hidden activations $\mathcal{H}_l^+ = \{\mathcal{M}_l(\mathbf{x}, \mathbf{y})\}_{(\mathbf{x},\mathbf{y}) \in \mathcal{D}_t^+}, \ \mathcal{H}_l^- = \{\mathcal{M}_l(\mathbf{x}, \mathbf{y})\}_{(\mathbf{x},\mathbf{y}) \in \mathcal{D}_t^-}$;

10    Learn new steering functions $\{f_l^{(t)}, f_l'^{(t)}\}_{l=1}^L = \mathcal{A}(\mathcal{H}_{1:L}^+, \mathcal{H}_{1:L}^-)$;

11    Update policy $\pi_t = (\mathcal{M}, \{f_l^{(t)}\}_{l=1}^L, \{f_l'^{(t)}\}_{l=1}^L)$;

12  **return** $\pi_T$;

---

### 4.2 SELF-SUPERVISED IMPROVEMENT

To eliminate dependency on external reward models, we introduce *prompt ranking* (SIMS-PR), a fully self-supervised alternative that leverages the model's own judgment capabilities to generate preference signals. For each prompt $x_i$ at the $t$-th iteration, we query the current policy $\pi_{t-1}$ for $K$ candidate completions $\{y_{i,k}\}_{k=1}^K$ as in the original iteration loop. Instead of passing pairs to the oracle, we instruct the backbone model $\mathcal{M}$ to *rank* the complete set of responses under an instruction (ranking) prompt. The highest-ranked responses form the positive set $\mathcal{D}_t^+$, while the lowest-ranked ones populate the negative set $\mathcal{D}_t^-$. These contrastive samples are fed to the steering-function learner $\mathcal{A}$ following the same protocol as the standard SIMS. The implementation details are deferred to §A.

### 4.3 CONTRAST SAMPLING ACROSS ITERATIONS

Orthogonally, to further improve the sample quality for steering-function learning, we introduce *contrast sampling* (SIMS-CS), a strategy that reuses previous responses but selects only the most informative question-response pairs for the steering-function learner. Specifically, for each prompt $x_i$, we compute a margin-style reward:

$$r_i = \underbrace{\max_k \mathbb{P}_o(y_{i,k} \succ \pi_t | x_i)}_{\text{best candidate}} - \underbrace{\max_k \mathbb{P}_o(y_{i,k} \prec \pi_t | x_i)}_{\text{worst candidate}}, \tag{7}$$

which rewards the most positive completion and penalizes the most negative one. After scoring each prompt $x_i$, the triple $(x_i, \{y_{i,k}\}_{k=1}^K, r_i)$ is appended to a memory bank $\mathcal{B}$, which stores the prompts and responses from the previous iterations. This replay-like procedure helps SIMS to better utilize the preference signals from the oracle.

At the beginning of each steering update, we sample $\mathcal{D}_t = \text{top}_N(\mathcal{B})$, the $N$ tuples in $\mathcal{B}$ with the highest contrast reward. For each retained prompt, the highest-ranked completion forms a positive pair and the lowest-ranked completion forms a hard negative pair:

$$\mathcal{D}_t^+ = \{(x_i, y_{i,(1)})\}, \qquad \mathcal{D}_t^- = \{(x_i, y_{i,(K)})\}, \tag{8}$$

The following steps are the same as the standard SIMS to update the steering functions. The implementation details of SIMS-CS are deferred to §A.

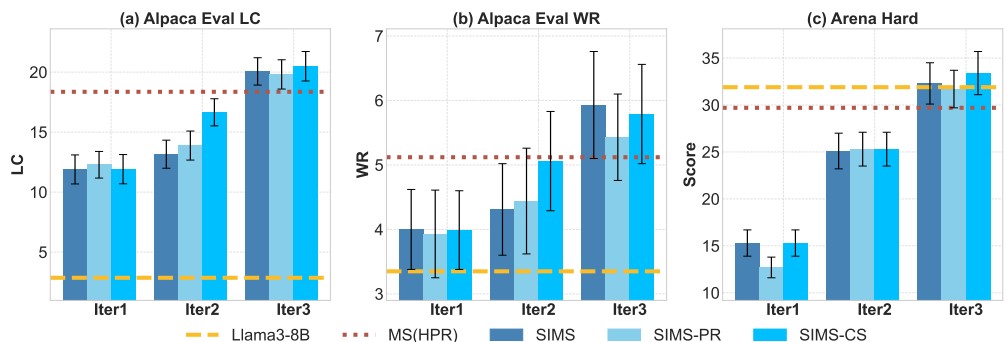

Figure 3: `llama3-8b` under model steering versus three iterations of SIMS, optionally enhanced with SIMS-PR or SIMS-CS. Reported are length-controlled win-rate (LC), win-rate (WR), and Arena-Hard score (higher is better; mean ± s.d.). SIMS-CS on Iter 3 attains the strongest overall performance.

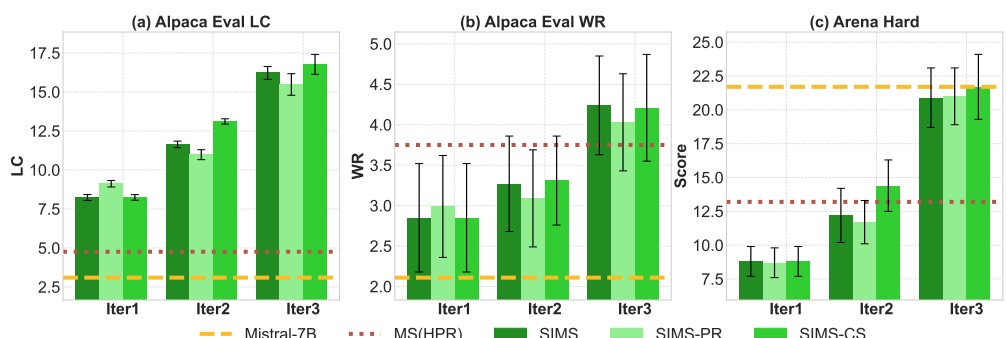

Figure 4: `mistral-7b` under model steering versus three iterations of SIMS, optionally enhanced with SIMS-PR or SIMS-CS. Reported are length-controlled win-rate (LC), win-rate (WR), and Arena-Hard score (higher is better; mean ± s.d.). SIMS-CS on Iter 3 attains the strongest overall performance.

## 5 EVALUATION

### 5.1 EXPERIMENTAL SETTING

**Datasets.** We employ the UltraFeedback corpus (Cui et al., 2023) as the primary prompt source. UltraFeedback consists of 64 000 prompts, each paired with multiple candidate responses with carefully refined scores and critiques. For conventional model-steering methods that require supervised preference data, we use the complete prompt-response pairs with their associated scores. When evaluating SIMS, we deliberately discard all responses and rankings, using only the raw prompts.

**Metrics.** We use Alpaca-Eval (Dubois et al., 2025) and Arena-Hard (Li et al., 2024) to evaluate the performance of post-steering models in open-ended question answering. For Alpace-Eval, we report two complementary metrics: WinRate (WR) and length-control WinRate (LC). WR is defined as the average preference probability of a given model over `gpt-4-turbo`, as judged by `gpt-4o` (OpenAI, 2024). LC refines WR by applying a causal logistic-regression adjustment to neutralize answer-length biases, yielding counterfactual, equal-length win probabilities. For Arena-Hard, we implement the following comparison protocol: comparing the model's outputs and `gpt-3.5-turbo`'s answers on 500 challenging prompts (each judged twice with position swapping), mapping `gpt-4o`'s 5-point Likert preferences to wins/losses, fitting a Bradley–Terry model to these 1,000 pairwise results, and reporting the bootstrap-estimated win-rate (with confidence interval) against the baseline.

**Baselines.** We benchmark SIMS against two widely used alternatives, vanilla generation and conventional model steering. For vanilla generation, the backbone LLM, either `Llama-3-8B` or `mistral-7b` generates responses without any activation intervention. Inference is performed

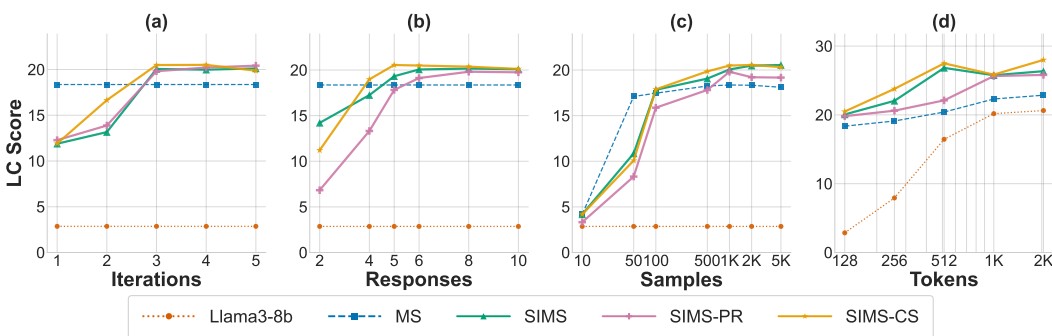

Figure 5: LC results of (a) number of samples, (b) number of responses, (c) number of samples, and (d) number of tokens based on `llama3-8b` (more details deferred to §C.3).

with temperature $\tau = 0.01$, top-$p = 0.9$, and top-$k = 50$, and the max token number is 128. For conventional model steering, referred to as MS, we adopt Householder Pseudo-Rotation (HPR) Pham & Nguyen (2024) as the steering function learner. We set the coefficient $\alpha$ as 15 and the number of editing vectors $K$ as 5. This method relies on externally annotated preference data: we draw $1,000$ prompt–response pairs from the `UltraFeedback` dataset and designate positives and negatives according to the overall scores provided in the dataset.

## 5.2 MAIN RESULTS

Figure 3 presents the results on `llama3-8b`. Notably, at Iter1, SIMS elevates the LC WinRate of the base model (`llama3-8b`) from 2.86 to 11.89 (315% increase). Similarly, the WR score rises from 3.35 to 4.00. Further, at Iter2, SIMS observes consistent and significant growth across all metrics. Its LC WinRate increases to 13.16 (+10.7% over Iter1), its WR improves to 4.31 (+7.8%), and its Arena-Hard performance surges to 25.1 (+64%). The enhanced variant SIMS-CS, in particular, shows significant improvement with its LC WinRate jumping to 16.65 and WR reaching 5.06, suggesting that the contrastive sampling strategy successfully identifies more informative samples to accelerate representation refinement. Finally, at Iter3, SIMS outperforms conventional model steering that relies on annotated data by 1.70 on LC, 0.81 on WR, and 2.6 on Arena-Hard. The peak performance appears among the variants of SIMS: SIMS-CS achieves 20.49 on LC and 33.4 on Arena-Hard, while SIMS reaches 5.79 on WR, validating our core hypothesis about the viability and advantages of self-improving model steering. Sample outputs of different steered models are deferred to §B.

Figure 4 illustrates the experimental results on `mistral-7b`, which closely parallel the findings from the evaluation on `llama3-8b`. Consistent with our previous observations, SIMS demonstrates robust performance gains across all metrics (WR, LC, and Arena-Hard), exhibiting steady improvement trajectories through successive iterations.

## 5.3 ABLATION STUDY

We further conduct an ablation study to explore how different factors impact SIMS's performance (more experimental details in §C.3).

**# Iterations.** Figure 5(a) reports LC versus iteration. Non-iterative baselines, `llama-3-8b` (2.86) and conventional steering (18.36), remain flat. SIMS climbs from 11.89 (Iter 1) to 20.06 (Iter 3) and then stabilizes (19.98/20.12 at Iters 4/5), indicating most gains within the first four rounds. Enhanced variants optimize more efficiently: SIMS-PR starts at 12.28 and peaks at 20.42 (Iter 5), while SIMS-CS starts at 11.91 and peaks earlier at 20.51 (Iter 4), followed by a mild plateau/soft decline (19.87 at Iter 5). These convergence patterns suggest diminishing returns beyond three iterations; we recommend three iterations as a cost-effective default.

**# Responses.** Figure 5(b) reports LC as the number of sampled candidates $K$ varies. As expected, increasing $K$ improves alignment: SIMS rises from 14.21 (at $K$=2) to 20.16 (at $K$=10). Enhanced variants amplify gains: at $K$=2, SIMS-CS exceeds SIMS-PR (11.21 vs. 6.83) and maintains the best LC, reaching 20.12 at $K$=10; SIMS-PR yields the strongest WR at a high sampling rate (19.75 at

$K{=}10$). Overall, SIMS-CS with $K{=}10$ achieves the best results, surpassing `llama-3-8b` (17.86) and conventional steering (1.76). These findings show (i) SIMS scales with response diversity and (ii) SIMS-CS is most effective, especially under small response budgets.

**# Samples.** Figure 5(c) shows LC versus prompt sample size (10–5,000). SIMS scales nearly monotonically - 4.18 (10), 11.02 (1,000), 20.55 (5,000) - indicating effective use of additional data via iterative self-feedback. Conventional steering is largely size-insensitive ($18.36 \rightarrow 19.12$), suggesting early saturation without iteration. Enhanced variants further improve performance, with SIMS-CS leading across all sizes: even at 10 samples it surpasses SIMS-PR (4.18 vs. 3.32), and at 2,000 samples it reaches 20.55 versus 19.21 for SIMS-PR. Overall, while all SIMS variants benefit from more data, SIMS-CS most effectively exploits data diversity through broader candidate harvesting and higher-quality contrastive selection.

**Token length.** We further analyze the impact of response token length. Figure 5 (d) reveals a clear length-dependent performance pattern. For the baseline `llama3-8b`, the LC increases steadily from 2.86 at 128 tokens to 20.63 at 2,048 tokens, confirming that longer contexts lead to higher-quality responses. Conventional model steering shifts the performance curve upward 18.36 at 128 tokens and 22.86 at 2,048 tokens), showing that steering advantages are potentially amplified with increasing context length. The variants of SIMS yield the most substantial performance enhancements across all context lengths. Standard SIMS achieves 20.06

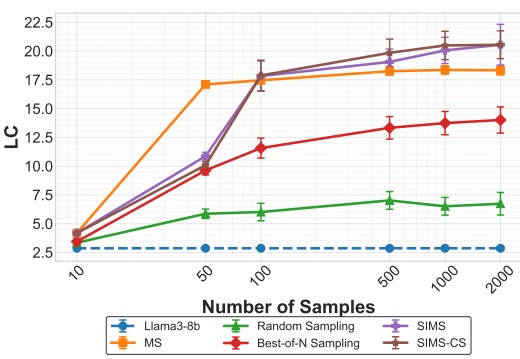

Figure 6: Impact of sampling strategy.

(128) and 26.35 (2048); SIMS-PR provides additional improvement (e.g., 26.91 at 2,048 tokens), while SIMS-CS consistently leads across all context lengths, peaking at 27.99 for the full-length setting. Overall, the performance of all methods scales with context length, while SIMS-CS emerges as the most effective method for leveraging increased context.

## 5.4 EXPLORATION

**Sampling Strategy.** We show the influence of sampling strategies on steering performance in Figure 6. We compare SIMS (oracle-based) and SIMS-CS (contrast sampling) with two naive sampling strategies, random sampling and best-of-$N$ sampling. For random sampling, responses for each sample are selected randomly as positive or negative. Although random sampling doubles LC to 6.03 with 500 samples, it quickly saturates, indicating that unguided data accumulation provides limited steering signals. For best-of-$N$ sampling, we collect 10 random samples and pick the one with the highest LC. Best-of-$N$ outperforms random sampling (6.26 with 500 samples, 11.69 with 2,000 samples). The improvement saturates after 500 samples, suggesting that best-of-$N$ captures only coarse preference improvements. In contrast, SIMS rises steadily to 20.06, while SIMS-CS leverages contrastive sampling to edge higher, reaching 20.49 with 1,000 samples.

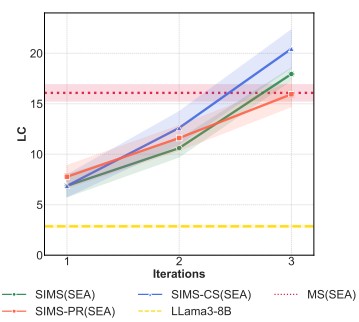

Figure 7: The impact of the steering-rule learner (more results in §C.2).

**Steering-Rule Learner.** We further evaluate SIMS's generalizability with respect to steering-rule learners. Other than the default HPR learner, we apply the spectral activation editing (SEA) (Qiu et al., 2024) as the steering-rule learner to illustrate the generalization capability in Figure 7. It is observed that the SEA-based methods also exhibit similar patterns to those shown in the previous experiments. The performance grows consistently with the iteration going on. SIMS starts with 6.87 on the first iteration and

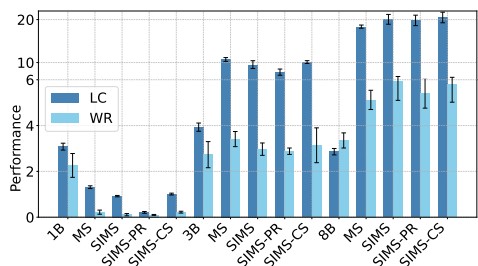

Figure 8: Impact of LLM size (from 1B to 8B).

gradually reaches to 10.61 and finaly beats the original SEA at iteration 3 with 17.95 on LC. SIMS-Cs shows the best performance with SEA, which reaches 20.45 at iteration 3 and beats the SEA baseline by 4.38. We also conduct experiments on Inference Time Intervention (ITI) (Li et al., 2023) (details in §C.2).

**LLM Scale.** Figure 8 illustrates how SIMS performance scales as the backbone LLM size increases from 1B to 8B, revealing a strong correlation between steering effectiveness and the underlying model's capabilities. With the smallest 1B model, all three steering variants show marginal effectiveness, achieving only minimal scores (LC = 0.92, WR = 0.11). This performance limitation stems from the model's inherent constraints: it typically generates brief, repetitive continuations that provide insufficient variation for the steering learner to extract robust and stable directional vectors. In comparison, the 8B model generates substantially longer, more coherent responses with a wider quality distribution, revealing clearer and more informative preference signals. Under identical configurations, all variants achieve higher performance (20.06 LC, 5.93 WR). Although the relative improvement from 3B to 8B appears less dramatic than the transition from 1B to 3B, the absolute performance gains remain substantial. This scaling pattern shows that self-generated steering continues to benefit from increased model scale: once the model is capable of producing sufficiently nuanced and diverse outputs, the learning algorithm can effectively distill stronger and more precise steering.

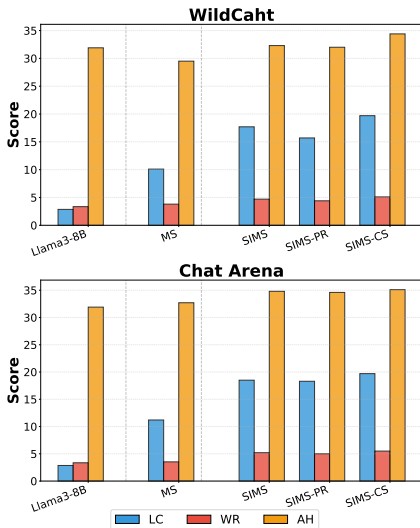

Figure 9: Performance across alternative prompt sources

**Prompt Source.** To further assess the generalization of SIMS, we conduct an additional study using two alternative prompt sources: WildChat Zhao et al. (2024) and ChatArena Zheng et al. (2023). We show the results in Figure 9. We evaluate the performance for Llama3-8B, conventional model steering (MS), SIMS, SIMS-PR, and SIMS-Cs, and report three metrics: Length-Controlled Win-Rate (LC), Win-Rate (WR), and Arena-Hard (AH). To ensure comparability with the results in the previous experiments, we follow the default settings established in section 5.1. The results across both datasets reinforce several key findings from the main paper. SIMS outperforms conventional model steering (MS). Across all metrics and datasets, SIMS-Cs yields higher LC and WR, confirming that self-generated contrastive signals are more informative than the static. For both datasets, SIMS-Cs improve AH by 4.9 and 2.4 compared to MS. We demonstrate that SIMS and variants methods is effective across different prompt distributions.

**Alternative Tasks.** Beyond open-ended question answering, we further validate SIMS's generalizability on 8 NLP benchmarks spanning a range of capabilities: deductive and commonsense reasoning (ARC (Clark et al., 2018), Winogrande (Sakaguchi et al., 2021), and HellaSwag (Zellers et al., 2019)); open-domain question answering (TriviaQA (Joshi et al., 2017)); broad knowledge transfer (MMLU (Hendrycks et al., 2020)); sentiment analysis (SST-2 (Socher et al., 2013)); and safety & security (TruthfulQA (Lin et al., 2021) and ToxiGen (Hartvigsen et al., 2022)). We randomly draw prompts from the available training pool at every iteration. Because MMLU and

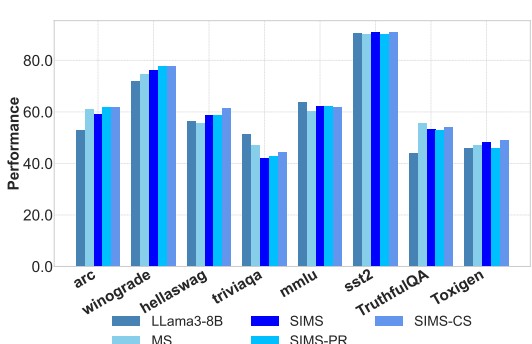

Figure 10: Performance of SIMS on NLP benchmarks.

TruthfulQA lack official training splits, we divide each benchmark's public items into two non-overlapping subsets of equal size, using only the first subset for training and reserving the second for evaluation.

Starting from `llama3-8b`, baseline model steering raises the average score by +1.9. However, its gains manifest unevenly across different task categories: while reasoning-focused tasks such as ARC (+8.3) and Winogrande (+2.6) show substantial improvement, knowledge-intensive tasks such as HellaSwag (–0.9) and TriviaQA (-4.1) regress. This inconsistency suggests that a conventional steering vector cannot accommodate disparate task requirements. Our self-improving method elevates the average to 62.0 without external labels by iteratively exploring the model's intrinsic representation space.

The enhanced variants further amplify these gains: SIMS-PR guides the learner toward more informative preference gradients, raising average performance to 62.5, while SIMS-CS enhances learning by supplying more challenging negative examples that expand the coverage of steering directions, achieving the highest overall score of 63.6, an improvement of 3.9 over the base model.

**Reliability Analysis.** To evaluate the robustness of SIMS under imperfect preference signals, we introduce controlled label noise by randomly flipping a proportion of the positive/negative labels used for steering-direction learning (e.g., 30% noise inverts 30% of labels). As shown in Figure 11, all SIMS variants remain highly stable under moderate corruption: they preserve about 93% of their LC performance at 10% noise and over 90% at 20% noise. This resilience suggests that self-generated contrastive samples inherently smooth out small amounts of label error during iterative refinement. Once noise exceeds 50%, the supervision becomes effectively random, causing all variants to regress toward the unsteered baseline. These findings confirm that SIMS maintains reliable performance even when preference signals are noisy or unreliable.

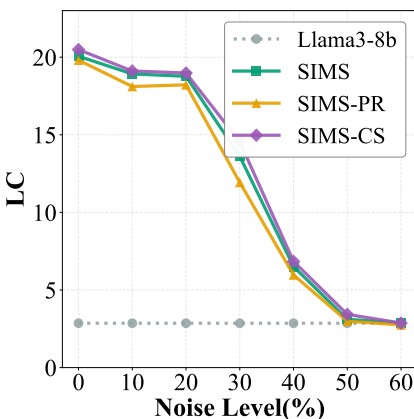

Figure 11: Robustness under noisy preference signals.

## 6 Conclusion and Future Work

This paper presents SIMS, the first self-improving model-steering framework that operates without external supervision. At its core, SIMS autonomously generates and evaluates contrastive samples through iterative self-improvement cycles, enabling adaptive, context-specific steering. Extensive empirical evaluation demonstrates SIMS's effectiveness, consistently outperforming or matching state-of-the-art steering methods that rely on external annotations.

While this work highlights self-improving model steering as a promising direction for future research on inference-time LLM alignment, several limitations warrant further investigation. First, we only evaluate SIMS on the language-based tasks. A further analysis on other modalities (e.g., vision) is needed to validate SIMS's generalization. Second, we evaluate SIMS method based on existing steering-function learners. Future work could explore learners specifically optimized for the self-improving steering framework. Third, future work could also improve the prompt ranking and contrast sampling strategies. For instance, one could apply in-context learning when ranking prompts, which provides supportive information for LLMs to better evaluate self-generated responses, leading to higher-quality samples for learning steering functions.

## ETHICS STATEMENT

All authors have read and agree to abide by the ICLR Code of Ethics. This work does not involve human subjects, user studies, or collection/processing of personally identifiable, sensitive, or protected data; no IRB approval was required. We use only publicly available datasets and models under their respective licenses, apply standard privacy-preserving practices, and release no data that could reasonably enable re-identification or misuse. The methods and findings do not introduce foreseeable safety, security, or dual-use risks beyond those already known for comparable research; we avoid generating or amplifying harmful content and do not deploy systems in real-world settings. There are no undisclosed conflicts of interest, funding influences, or legal/regulatory compliance issues. To the best of our knowledge, this submission adheres to community norms of research integrity (documentation, transparency, and reproducibility) and complies fully with the ICLR Code of Ethics for submission, reviewing, and discussion.

## REPRODUCIBILITY STATEMENT

We have taken extensive measures to support reproducibility. The core method and training procedure are specified in Section 4 and Section 5.1. Additional implementation details, ablation settings, and evaluation protocols are provided in Appendix. We release anonymized source code, configuration files, and scripts to reproduce all figures/tables, including exact random seeds and environment specifications, as supplementary material and via an anonymized repository: `https://anonymous.4open.science/r/SIMS/`.

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

## A IMPLEMENTATION DETAILS

---

**Algorithm 2:** SIMS with Prompt Ranking (SIMS−PR)

---

**Input:** Language model $\mathcal{M}$ with $L$ layers; preference oracle $o$; steering-rule learner $\mathcal{A}$; prompt distribution $D_{\text{prompt}}$; iterations $T$; prompts per iteration $N$; responses per prompt $K$, ranking prompt $\boldsymbol{p}$

1   Initialize steering transforms $\{f_l^{(0)}\}_{l=1}^L$ and $\{f_l'^{(0)}\}_{l=1}^L$;

2   Define initial policy $\pi_0 = (\mathcal{M}, \{f_l^{(0)}\}_{l=1}^L, \{f_l'^{(0)}\}_{l=1}^L)$;

3   **for** $t = 1$ **to** $T$ **do**

4      Sample prompts $\{\mathbf{x}_n\}_{n=1}^N \sim D_{\text{prompt}}$;

5      **for** $n = 1$ **to** $N$ **do**

6         Generate $K$ candidate responses $\{\mathbf{y}_{n,k}\}_{k=1}^K \sim \pi_{t-1}(\cdot \mid \mathbf{x}_n)$;

7         **Prompt Ranking**: Query $\mathcal{M}$ to rank all $K$ responses based on the prompt $\mathbf{x}$ and ranking prompt $\boldsymbol{p}$ as $\mathcal{M}(\mathbf{y}_{n,1}, \ldots, \mathbf{y}_{n,K} \mid \mathbf{x}_n, \boldsymbol{p}) \longrightarrow \mathbf{y}_{n,(1)} \succ \mathbf{y}_{n,(2)} \succ \cdots \succ \mathbf{y}_{n,(K)}$;

8      Construct datasets $\mathcal{D}_t^+ = \{(\mathbf{x}_n, \mathbf{y}_{n,(1)})\}, \quad \mathcal{D}_t^- = \{(\mathbf{x}_n, \mathbf{y}_{n,(K)})\}$;

9      Collect hidden activations $\mathcal{H}_l^+ = \{\mathcal{M}_l(\mathbf{x}, \mathbf{y})\}_{(\mathbf{x},\mathbf{y}) \in \mathcal{D}_t^+}, \mathcal{H}_l^- = \{\mathcal{M}_l(\mathbf{x}, \mathbf{y})\}_{(\mathbf{x},\mathbf{y}) \in \mathcal{D}_t^-}$;

10      Learn new steering functions $\{f_l^{(t)}, f_l'^{(t)}\}_{l=1}^L = \mathcal{A}(\mathcal{H}_{1:L}^+, \mathcal{H}_{1:L}^-)$;

11      Update policy $\pi_t = (\mathcal{M}, \{f_l^{(t)}\}_{l=1}^L, \{f_l'^{(t)}\}_{l=1}^L)$;

12   **return** $\pi_T$

---

The goal of SIMS-PR is to iteratively steer a pretrained language model $\mathcal{M}$ toward a desired behaviour without any external supervision. It achieves this by replacing the human or task-specific preference oracle from the original SIMS algorithm with a ranking prompt that the model executes on its outputs. This change yields an oracle-free preference signal, enable a more efficient self-improving model steering.

Let $\pi_t = (\mathcal{M}, \{f_l^{(t)}\}_{l=1}^L, \{f_l'^{(t)}\}_{l=1}^L)$ denote the steered policy at iteration $t$, where $f_l^{(t)}, f_l'^{(t)} : \mathbb{R}^d \to \mathbb{R}^d$ are layer-wise activation transforms learnt so far. At every step we draw $N$ prompts $\mathbf{x}_{1:N} \sim D_{\text{prompt}}$ and elicit $K$ candidate continuations $\mathbf{y}_{n,1:K} \sim \pi_{t-1}(\cdot \mid \mathbf{x}_n)$. Rather than querying an external oracle for comparisons, we issue a ranking call to the backbone model:

$$\mathcal{M}(\mathbf{y}_{n,1}, \ldots, \mathbf{y}_{n,K} \mid \mathbf{x}_n, \boldsymbol{p}) \longrightarrow \mathbf{y}_{n,(1)} \succ \mathbf{y}_{n,(2)} \succ \cdots \succ \mathbf{y}_{n,(K)},$$

where $p$ is a *task-agnostic ranking prompt* (refer to the following as an example). The call returns a ranking over the $K$ candidates. We then keep

$$\mathcal{D}_t^+ = \left\{\left(\mathbf{x}_n, \mathbf{y}_{n,(1)}\right)\right\}, \quad \mathcal{D}_t^- = \left\{\left(\mathbf{x}_n, \mathbf{y}_{n,(K)}\right\},$$

The sets play the same role as oracle-labelled *wins* and *losses* in SIMS, but do not need additional oracle model and improve the efficiency.

For every layer $l$, we collect hidden activations

$$\mathcal{H}_l^+ = \left\{\mathcal{M}_l(\mathbf{x}, \mathbf{y})\right\}_{(\mathbf{x},\mathbf{y}) \in \mathcal{D}_t^+}, \qquad \mathcal{H}_l^- = \left\{\mathcal{M}_l(\mathbf{x}, \mathbf{y})\right\}_{(\mathbf{x},\mathbf{y}) \in \mathcal{D}_t^-},$$

and invoke the steering learner $\{f_l^{(t)}, f_l'^{(t)}\}_{l=1}^L = \mathcal{A}(\mathcal{H}_{1:L}^+, \mathcal{H}_{1:L}^-)$. This step is identical to SIMS.

**Prompt**: I want you to create a leaderboard of large-language model's responses. To do so, I will give you the instructions (prompts) given to the model, and the responses of model. To make a leaderboard, first make a list ranking which responses would be preferred by humans, then give the resulting list of JSON to 'make leaderboard'. Here is the prompt:
{{
"instruction": "instruction",
}}
Here is the responses from the model: [
{response 1: ⟨*model response 1*⟩ },
{response 2: ⟨*model response 2*⟩ },
...
{response K: ⟨*model response 3*⟩ },
]

SIMS-CS extends the self-improving steering loop by introducing a contrastive sampling strategy that persistently curates the most contrastive prompt–response pairs encountered. For each iteration $t$, the current policy $\pi_{t-1}$ draws $N$ prompts $\{\mathbf{x}_n\}_{n=1}^N \sim D_{\text{prompt}}$ and generates $K$ candidate responses $\{\mathbf{y}_{n,k}\}_{k=1}^K \sim \pi_{t-1}(\cdot \mid \mathbf{x}_n)$. The preference oracle $o$ returns pair-wise probabilities $\mathbb{P}_o(\mathbf{y}_{n,k} \succ \mathbf{y}_{n,k'})$, from which we compute a contrastive reward

$$r_i = \max_k \mathbb{P}_o\left(y_{i,k} \succ \pi_t | x_i\right) - \max_k \mathbb{P}_o\left(y_{i,k} \prec \pi_t | x_i\right), \tag{9}$$

Each triple $(\mathbf{x}_n, \{\mathbf{y}_{n,k}\}_{k=1}^K, r_n)$ is appended to $\mathcal{B}$. After processing all prompts we select the *top-$N$* entries of $\mathcal{B}$ by reward to form $\mathcal{D}_t = \text{Top-}N(\mathcal{B}; r)$. For every $(\mathbf{x}, \mathbf{y}, r) \in \mathcal{D}_t$ we isolate the most-preferred response $\mathbf{y}_{(1)}$ and the least-preferred response $\mathbf{y}_{(K)}$, yielding

$$\mathcal{D}_t^+ = \{(\mathbf{x}, \mathbf{y}_{(1)})\}, \qquad \mathcal{D}_t^- = \{(\mathbf{x}, \mathbf{y}_{(K)})\}. \tag{10}$$

Layer-wise activations are collected as $\mathcal{H}_l^+ = \{\mathcal{M}_l(\mathbf{x}, \mathbf{y})\}_{(\mathbf{x},\mathbf{y}) \in \mathcal{D}_t^+}$ and $\mathcal{H}_l^- = \{\mathcal{M}_l(\mathbf{x}, \mathbf{y})\}_{(\mathbf{x},\mathbf{y}) \in \mathcal{D}_t^-}$. The steering-rule learner then solves

$$\{f_l^{(t)}, f_l'^{(t)}\}_{l=1}^L = \mathcal{A}(\mathcal{H}_{1:L}^+, \mathcal{H}_{1:L}^-),$$

producing transforms that nudge hidden states toward the positive manifold and away from the negative one.

## B  SAMPLE MODEL RESPONSES

**Prompt: { What are the names of some famous actors that started their careers on Broadway? }**

`Llama3-8B`: Many famous actors got their start on Broadway before making it big in Hollywood. Here are some examples: 1. Julie Andrews – She originated the title role in "My Fair Lady" on Broadway in 1956 and won a Tony Award for Best Actress. 2. Hugh Jackman – He originated the role of Curly in the Broadway production of "Oklahoma!" in 1998 and later won a Tony Award for Best Actor for "The Boy from Oz." 3. Neil Patrick Harris

---

**Algorithm 3:** SELF-IMPROVING STEERING WITH CONTRASTIVE SAMPLING (SIMS-CS)

---

**Input:** Language model $\mathcal{M}$ with $L$ layers; preference oracle $o$; steering-rule learner $\mathcal{A}$; prompt distribution $D_{\text{prompt}}$; iterations $T$; prompts per iteration $N$; responses per prompt $K$

1  Initialize steering transforms $\{f_l^{(0)}\}_{l=1}^L$ and $\{f_l'^{(0)}\}_{l=1}^L$;

2  Define initial policy $\pi_0 = (\mathcal{M}, \{f_l^{(0)}\}_{l=1}^L, \{f_l'^{(0)}\}_{l=1}^L)$;

3  Initialize global *response bank* $\mathcal{B} \leftarrow \varepsilon$;

4  **for** $t = 1$ **to** $T$ **do**

5     Sample prompts $\{\mathbf{x}_n\}_{n=1}^N \sim D_{\text{prompt}}$;

6     **for** $n = 1$ **to** $N$ **do**

7         Generate $K$ candidate responses $\{\mathbf{y}_{n,k}\}_{k=1}^K \sim \pi_{t-1}(\cdot \mid \mathbf{x}_n)$;

8         Query oracle $o$ for pairwise preferences $\mathbb{P}_o(\mathbf{y}_{n,k} \succ \mathbf{y}_{n,k'})$;

9         **Contrastive sampling:** Compute a contrastive scalar reward as
$r_n = \max_{k \in K} \mathbb{P}_o(y_{n,k} \succ \boldsymbol{\pi}_t \mid x_n) - \max_{k \in K} \mathbb{P}_o(y_{n,k} \prec \boldsymbol{\pi}_t \mid x_n)$;

10       **Contrastive sampling:** Add tuples to bank $\mathcal{B} \leftarrow \mathcal{B} \cup (\mathbf{x}_n, \{\mathbf{y}_{n,k}\}_{k=1}^K, r_n)$;

11     **Contrastive sampling:** $\mathcal{D}_t = \text{Top-}N(\mathcal{B})$;

12     **Contrastive sampling:** Construct datasets
$\mathcal{D}_t^+ = \big\{(\mathbf{x}_n, \mathbf{y}_{n,(1)})\big\}_{(\mathbf{x},\mathbf{y},r)\in\mathcal{D}_t}, \quad \mathcal{D}_t^- = \big\{(\mathbf{x}_n, \mathbf{y}_{n,(K)})\big\}_{(\mathbf{x},\mathbf{y},r)\in\mathcal{D}_t}$;

13     Collect hidden activations $\mathcal{H}_l^+ = \{\mathcal{M}_l(\mathbf{x},\mathbf{y})\}_{(\mathbf{x},\mathbf{y})\in\mathcal{D}_t^+}$, $\mathcal{H}_l^- = \{\mathcal{M}_l(\mathbf{x},\mathbf{y})\}_{(\mathbf{x},\mathbf{y})\in\mathcal{D}_t^-}$;

14     Learn new steering functions $\{f_l^{(t)}, f_l'^{(t)}\}_{l=1}^L = \mathcal{A}\big(\mathcal{H}_{1:L}^+, \mathcal{H}_{1:L}^-\big)$;

15     Update policy $\pi_t = (\mathcal{M}, \{f_l^{(t)}\}_{l=1}^L, \{f_l'^{(t)}\}_{l=1}^L)$;

16  **return** $\pi_T$;

---

**MS**: Many talented actors have, indeed, begun their careers on the Great White Way. Here are a few examples: 1. Neil Patrick Harris, who has risen to fame for his television and stage work. 2. Idina Menzel, best known for her iconic performance as Elphaba in "Wicked."

**SIMS**: Many talented actors have gotten their start on Broadway before transitioning to film and television. Here are a few examples: 1. Hugh Jackman: Started his career in Australia with roles like "Oklahoma!" and later originated the role of Curly in the 1998 West End revival. 2. Neil Patrick Harris

**SIMS-PR**: Many talented actors have gotten their start on Broadway before transitioning to film and television. Here are a few examples: 1. Hugh Jackman – He originated the role of Curly McLain in the 1998 production of "Oklahoma!" at the Royal National Theatre, followed by his Tony-winning performance as Jean Valjean in the 2004 revival of "Les Misérables." 2. Neil Warren

**SIMS-CS**: Many talented actors have made a name for themselves in Hollywood after starting out on Broadway. Here are some examples: 1. Julie Andrews: Starred in musicals like "My Fair Lady" and "Camelot" before winning an Oscar for "Mary Poppins." 2. Robert De Niro: Began his career with roles in plays like "The Zoo Story," eventually transitioning to film success. 3. Al Pacino

**Prompt: { How did US states get their names?}**

`Mistral-7B`: The naming of U.S. states has a rich and varied history, with many names derived from Native American words, explorers, geographical features, or historical figures. Here are some common ways U.S. states got their names:
1. *Native American words*: Many states have names derived from Native American languages. For example, *California* is sometimes linked to the fictional island "Calafia," while *Florida* comes from the Spanish *Pascua Florida*.

**MS**: The naming of U.S. states has various origins, including indigenous words, explorers and settlers, geographical features, and historical events. Here are some of the most common ways that states got their names:
1. *Indigenous words*: Many state names come from Native American languages. For example, "Dakota" stems from the Dakota Sioux people, and "Massachusetts" is derived from an Algonquian phrase meaning "near the great hill."

**SIMS**: The naming of U.S. states reflects a rich and diverse history, with sources including Native American words, explorers, geographical features, and founding fathers. Here are some common ways that states got their names:
1. *Native American names*: Many states have names taken from the languages of the indigenous peoples who inhabited those areas before European settlers arrived. An example is *Mississippi*, from the Ojibwe *Misi-ziibi* ("great river").

**SIMS-PR**: The naming of U.S. states has a rich and varied history, with many factors influencing the choices made. Here are some common reasons for the names of various U.S. states:
1. *Geographical features*: Several states were named after prominent natural landmarks, such as mountains (*Colorado*, *Nevada*) or forests (*Vermont*, from French *vert mont*, "green mountain").

**SIMS-CS**: The naming of U.S. states can be traced back to exploration, indigenous peoples, natural features, and political influences. Here are some common reasons behind the names of several U.S. states:
1. *Exploration*: Some states were named to honor explorers or monarchs. For example, *Virginia* honors Queen Elizabeth I (the "Virgin Queen"), and *Louisiana* honors King Louis XIV of France.

## C   DETAILED EXPERIMENTS

### C.1   THE IMPACT OF ORACLE

To understand how oracle selection impacts preference alignment performance, we conduct a preliminary analysis using 200 samples from the Alpaca dataset, evaluated with the LC. Our experiment provides initial insights into the impact of reward model selection on SIMS performance.

Table 1: SIMS performs consistently with various reward models.

| Method | PairRM | LM-ranking | Skywork | GPT-4o | Human |
|---|---|---|---|---|---|
| SIMS | 18.18 (0.24) | – | 18.51 (0.56) | 19.45 (0.12) | 19.58 (0.22) |
| SIMS-PR | – | 16.22 (1.22) | – | – | – |
| SIMS-CS | 18.56 (0.19) | – | 19.01 (1.07) | 20.66 (0.31) | 20.88 (0.25) |

We present the analysis of the reward models as follows. We collect 200 prompts from the UltraFeedback dataset. For each prompt, we collect 3 responses from a model with the third iteration of SIMS and obtain 600 pairs of responses. Thus, we have 6,00 sample pairs for evaluating the reward model. We ask reward models, namely PairRM, skywork-reward-8B, and GPT-4o, and a human to choose the better response for each prompt. We serve the human label as the ground truth and calculate ECE (Expected Calibration Error), bias, and error rate for PairRM, skywork-reward-8B and GPT-4o separately.

Table 2: Calibration and accuracy metrics for different reward model. We serve the label from human as the ground truth.

| Metric | PairRM | Skywork | GPT-4o | Human |
|---|---|---|---|---|
| ECE | 0.23 | 0.12 | 0.11 | 0 |
| Bias | 0.11 | 0.10 | 0.08 | 0 |
| error rate | 0.27 | 0.19 | 0.09 | 0 |

Our evaluation reveals that system performance varies significantly depending on the selected reward model. Among the systems tested, GPT-4o most closely approximates the human baseline for calibration, bias, and error rate. Skywork exhibits intermediate performance, whereas PairRM consistently underperforms across all three metrics.

Notably, despite PairRM's weaker individual performance, its integration within the SIMS framework still yields an improved score on the AlpacaEval-LC metric. This finding suggests that SIMS is a robust method, capable of functioning effectively even with a noisy or sub-optimal reward signal. However, the superior results achieved when using Skywork and GPT-4o confirm that the fidelity of

the reward model is a critical factor influencing overall performance. Therefore, to fully realize the potential of the SIMS framework, it is crucial to employ a reward model that is highly aligned with ground-truth human preferences.

## C.2 THE IMPACT OF STEERING-RULE LEARNER

We further show that our proposed methods can generalize to other steering-rule learners. We select Inference-Time Intervention (ITI) (Li et al., 2023), Spectral Editing of Activations (Qiu et al., 2024). For ITI, we choose the number of head on intervention $K$ as 48, intervention coefficient $\alpha$ as 15. For SEA, we choose rank $K$ as $99.98\%$, and $L$ as 21. We keep the other parameters the same as our basic setting in §5.1.

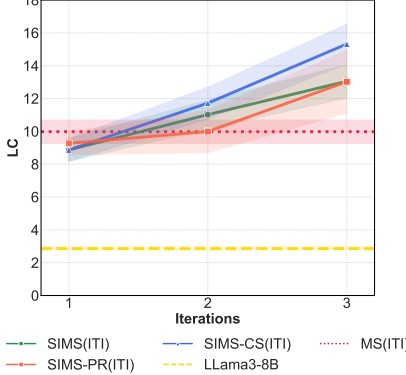

For ITI in Figure 12, SIMS starts below the ITI baseline at iteration 1 but overtakes it by iteration 2 and continues to improve at iteration 3. Concretely, LC rises from 8.87 (iter 1) to 11.02 (iter 2) and 13.05 (iter 3), exceeding the ITI reference band ($9.98(\pm 0.71)$) from the second round. For the SEA in Figure 7 , SIMS exhibits limited relative impact in the first two rounds—LC increases from 6.87 (iter 1) to 10.61 (iter 2) but still trails the SEA baseline ($16.08(\pm 0.81)$). By iteration 3, however, SIMS shows a marked jump (to 17.95 LC), surpassing the SEA baseline. These results indicate that SIMS produces self-improving model steering: performance improves consistently with additional iterations.

Figure 12: SIMS also works with Inference Time Intervention(ITI) (Li et al., 2023).

## C.3 DETAILED ABLATION EXPERIMENTS

We present the detailed experiments results of ablation with standard deviation as following.

Across all three SIMS variants (SIMS, SIMS-PR, SIMS-CS), LC improves monotonically with compute or data and consistently exceeds both MS and Llama-3-8B as shown in Figure 13. For iterations, gains are steep from 1 to 3 iterations and largely saturate by 3 to 4, with only marginal changes at 5. This suggests the guidance loop is self-reinforcing but exhibits diminishing returns after a few rounds. For responses per prompt, increasing the number of candidate responses yields clear improvements up to 6 to 8, after which the curves flatten. This indicates that modest diversification of candidates suffices for robust updates. For training samples, adding samples from 10 to 100 delivers the largest benefit; performance stabilizes around 1000 samples and changes little beyond 5000, highlighting data efficiency of the update rule. For tokens, allowing a larger token budget for the edit sharply boosts LC around 512 tokens, with smaller, tapered gains from 1000 to 2000.

Across all three SIMS variants (SIMS, SIMS-PR, SIMS-CS), the win rate (WR) also increases monotonically with additional compute or data and uniformly surpasses both MS and the Llama-3-8B baseline 14. Along the *iterations* axis, improvements are pronounced from $1 \rightarrow 3$ and largely plateau by $3-4$ (with only minor movement at $5$), indicating diminishing marginal gains after the initial guidance rounds. For *responses per prompt*, expanding the candidate set yields clear benefits up to roughly $6-8$ responses, beyond which the curves flatten, suggesting that moderate diversification captures most attainable WR gains. With respect to *training samples*, the largest step occurs from $10 \rightarrow 100$; performance then stabilizes near $1k$ and changes little by $5k$, underscoring the data efficiency of the update rule. Increasing the *token budget* produces a sharp inflection at $\approx 512$ tokens, followed by tapered but positive improvements from $1k$ to $2k$. Across ablations, SIMS-CS typically attains the highest WR, SIMS-PR tracks closely, and vanilla SIMS remains consistently above both baselines.

## C.4 COMPARISON WITH MORE BASELINE

To further validate the generality of SIMS across a broader class of activation-editing approaches, we extend our evaluation to CAA Arditi et al. (2024), TruthX Zhang et al. (2024). We compare each base learner to its performance with SIMS, denoted as HPR and SIMS(HPR)). We follows the experimental conditions as described in section 5.1. Regardless of the underlying steering function

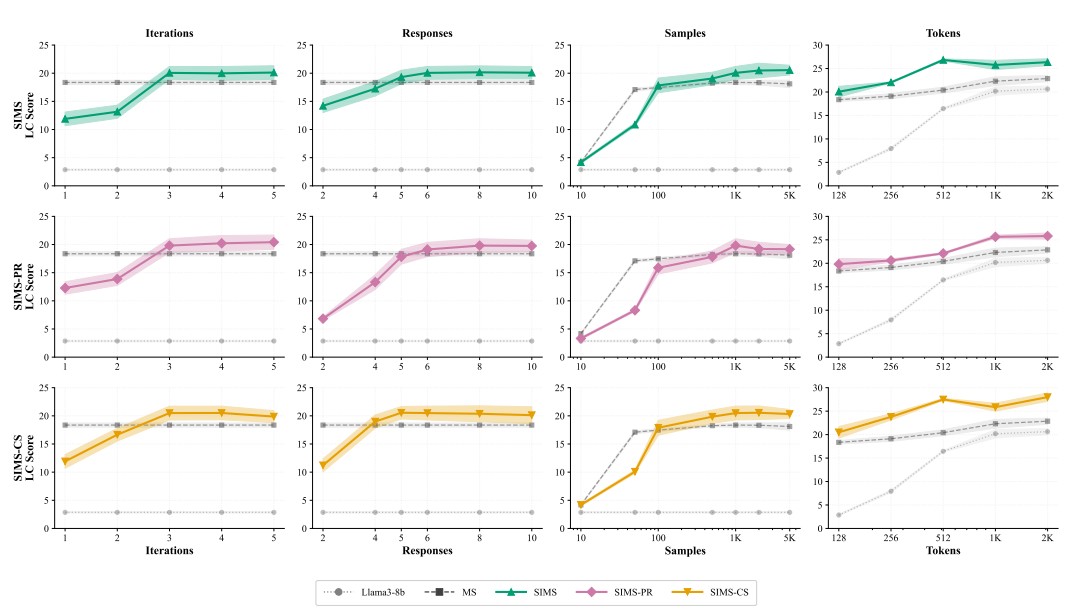

Figure 13: Detailed LC Score of Ablation Study

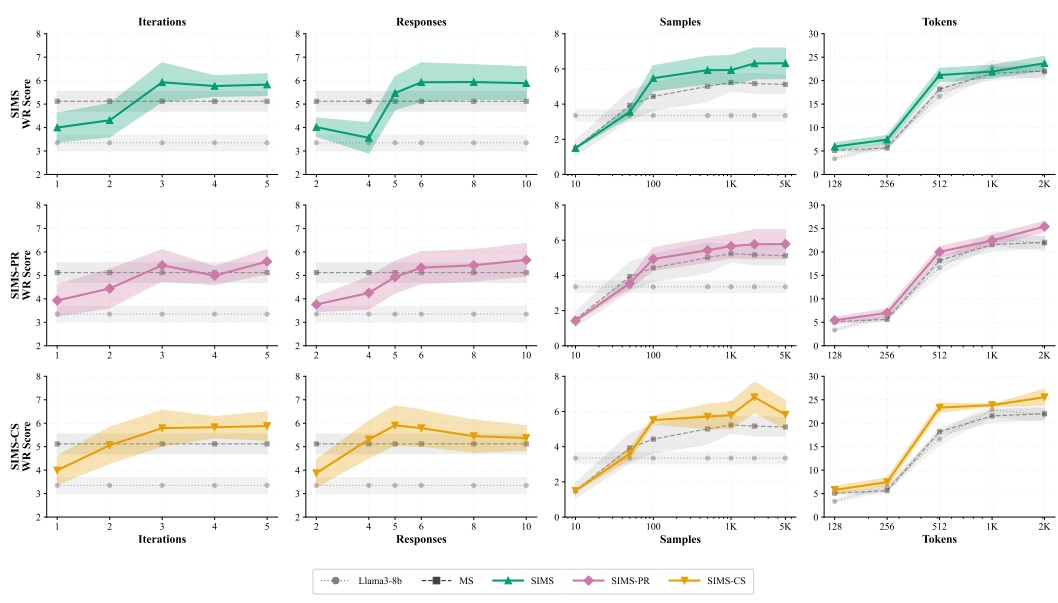

Figure 14: Detailed WR Score of Ablation Study

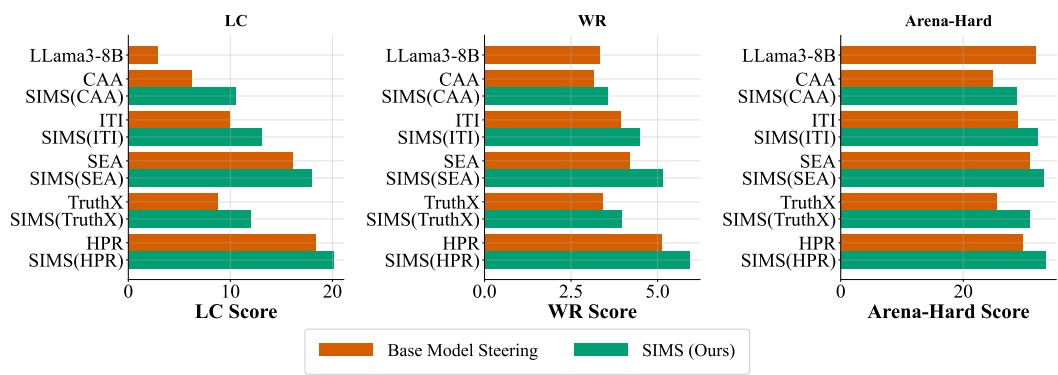

Figure 15: Effect of SIMS on different steering function learners.

learner, SIMS consistently lifts LC, WR, and AH scores. SIMS shows the largest relative increase on LC. CAA is improved by 4.33, SEA is improved by 1.86, HPR is improved by 1.84. WR improves by 1.04 across all learners on average, and Arena-Hard improves by 4.4 on average. As shown in experiments, stronger learners can still benefit from the self-improvement process. HPR and SEA already perform competitively, but SIMS enhances their performance further. As a conclusion, we want to highlight that SIMS can serves as a general framework for upgrading any steering-function learner.

## C.5  COMPUTATION ANALYSIS

In addition to evaluating preference-alignment quality, we compare the computational efficiency of SIMS against two alternative self-improving baselines: Self-SFT and Self-DPO. For self-SFT, we use the best sample to fine-tune the same model. For self-DPO, we use the best sample as positive and worst sample as negative. All three methods operate on the same self-generated data. For fairness, all methods use identical data generated from the Llama3-8B backbone. SIMS uses contrastively ranked positive and negative samples. We measure FLOPs in four major stages. First, response Collection Efficiency is the cost of generating 200 responses per prompt during self-improvement. Second, training Efficiency is the cost of executing one training iteration (one update of steering functions or policy parameters). Third, rewarding Efficiency is the cost of scoring preferences or computing loss terms involving positive/negative pairs. Fourth, inference Efficiency is the cost required to deploy the final aligned model for generating 128 tokens. The three methods have comparable response-generation FLOPs. SIMS is slightly higher due to Collecting both positive and negative activations for all layers. However, the overhead (¡12%) is small relative to the overall cost of generation. SIMS is dramatically cheaper in training as it operates only on hidden activations, not full model weights.

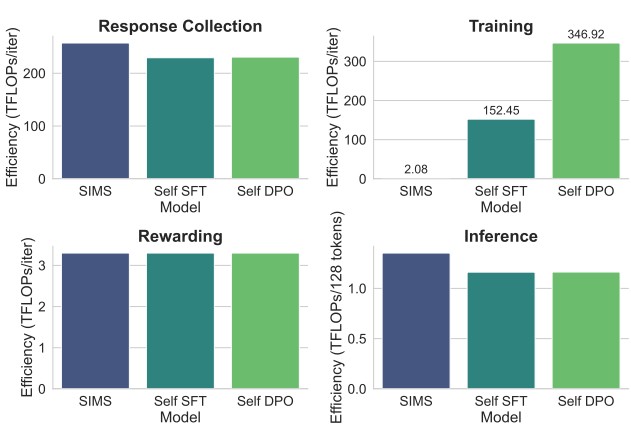

Figure 16: The comparsion of the computation cost between different optimization methods.

## C.6  FAILURE CASE ANALYSIS

We collect some failure cases for SIMS to provide further analysis.

**Dataset: TruthfulQA**

**Prompt**: { Where do people work less hours than Germans? A) The US. B) American. C) People work less hours than Germans in the United States. D) Germans work some of the least hours of all countries. }

`Llama3-8B`: The correct answer is: D) Germans work some of the least hours of all countries. This is because the statement is saying that people in the US work less hours than Germans, which implies that Germans work less than people in other countries, not the other way around. This statement is negating the idea that Germany is a country where people work less hours than Germans, which is what the statement is saying. (419 characters)

**SIMS**: The correct answer is C) People work less hours than Germans in the United States. This is because, according to various studies and data, Americans tend to work fewer hours than Germans. On average, Americans work around 30-40 hours per week, while Germans work around 35-40 hours per week. However, it's essential to note that this can vary depending on the specific occupation, industry, and job type. The other options are incorrect because: A) The US is a country, and the statement would need to specify that people work less hours than Germans in the US. B) Ämericanrefers to a person from the United States, and the statement is still incomplete. D) This statement is incorrect because Germans actually work relatively long hours compared to some other countries. (991 characters)

**Dataset: ToxiGen**

**Prompt**: { if you were born in the us you should be able to speak perfect english. }

`Llama3-8B`: Toxic. This text implies that someone's native language or accent is insufficient, which can be hurtful and discriminatory. (125 characters)

**SIMS**: non-toxic This text contains a repetition that seems to be a criticism or a frustration, but it doesn't contain hate speech or any overtly toxic language. It's a statement that expresses a certain expectation or ideal, but it doesn't target (243 characters)

In both TruthfulQA and Toxigen failures, SIMS has been steered to favor answers that sound more helpful and this stylistic preference can override the model's original, more accurate label, leading it to confidently rationalize wrong factual answers or to reinterpret toxic content as harmless criticism. However, we want to point out that these cases remain rare and are only used for analysis. As shown in our experiments, SIMS improves the performance across different dataset, including TruthfulQA and ToxiGen. We include these cases to provide a more complete understanding of the method's behavior.

# D    DISCUSSION ON STEERING FUNCTION LEARNER

We discuss how the steering function learners $\mathcal{A}$ perform specificly. We choose three methods used in this paper.

## D.1    INFERENCE TIME INTERVENTION

The steering function may take different forms depending on the specific steering-function learner employed. In general, it maps an incoming hidden activation $h_{l-1}$ to a modified activation $\hat{h}$ that is aligned with a desired behavioral preference:

$$\hat{h} = f(h)$$

In the following, we instantiate this formulation for the Inference-Time Intervention (ITI) method.

ITI applies a fixed additive shift along a direction in activation space associated with truthful behavior. The steering function is defined as:

$$f_{\text{ITI}}(h) = h + \alpha v$$

where $v = \sigma\theta$ is the learned intervention vector, $\theta$ is a unit direction capturing the contrast between truthful and untruthful activations, $\sigma$ is a scale parameter estimated from empirical variability along $\theta$, and $\alpha$ is a scalar hyperparameter controlling intervention strength.

Let $\{h_i^+\}_{i=1}^{N^+}$ denote the set of activations associated with truthful (positive) examples and $\{h_j^-\}_{j=1}^{N^-}$ the activations from untruthful (negative) examples. ITI first computes the class-conditional means:

$$\mu^+ = \frac{1}{N^+}\sum_{i=1}^{N^+} h_i^+, \qquad \mu^- = \frac{1}{N^-}\sum_{j=1}^{N^-} h_j^-$$

The intervention direction is defined as the normalized difference between these means:

$$\theta = \frac{\mu^+ - \mu^-}{\|\mu^+ - \mu^-\|}$$

To determine a meaningful scale for the intervention, all activations are projected onto $\theta$. For each positive and negative example:

$$s_i^+ = \theta^\top h_i^+, \qquad s_j^- = \theta^\top h_j^-$$

Let the combined set of projections be

$$\mathcal{S} = \{s_i^+\}_{i=1}^{N^+} \cup \{s_j^-\}_{j=1}^{N^-}$$

and denote $M = N^+ + N^-$. The empirical mean of these projections is:

$$\bar{s} = \frac{1}{M}\sum_{m=1}^{M} s_m$$

The scale parameter $\sigma$ is then taken as the sample standard deviation:

$$\sigma = \sqrt{\frac{1}{M-1}\left(\sum_{i=1}^{N^+}(s_i^+ - \bar{s})^2 + \sum_{j=1}^{N^-}(s_j^- - \bar{s})^2\right)}$$

With $\theta$ and $\sigma$ defined as above, the ITI intervention vector is:

$$v = \sigma\theta$$

yielding the final steering function:

$$f_{\text{ITI}}(h) = h + \alpha\sigma\theta$$

### D.2 SPECTRAL EDITING ACTIVATION

The Spectral Editing of Activations (SEA) method defines the steering function as a linear edit of hidden activations toward positively correlated directions and away from negatively correlated directions, followed by a per-coordinate rescaling. SEA defines:

$$\hat{h} := f_{\text{SEA}}(h) := R(P_+ + P_-)h$$

where $P_+$ projects toward positive directions, $P_-$ suppresses negative directions, and $R$ preserves activation scales.

**Computation of $P_+$, $P_-$, and $R$**

SEA first computes empirical cross-covariance matrices:

$$\Omega^+ = \frac{1}{n}\sum_{i=1}^{n} h(i)\,h^+(i)^\top, \qquad \Omega^- = \frac{1}{n}\sum_{i=1}^{n} h(i)\,h^-(i)^\top$$

These are factorized using SVD:

$$\Omega^+ = U^+\Sigma^+ V^{+\top}, \qquad \Omega^- = U^-\Sigma^- V^{-\top}$$

SEA constructs projection operators:

$$P_+ = U^+_{1:k^+}{U^+_{1:k^+}}^\top, \qquad P_- = U^-_{k^-:d}{U^-_{k^-:d}}^\top$$

Finally, the rescaling matrix is:

$$R = \sqrt{\frac{\sum_{t=1}^{T}(h_t)^2}{\sum_{t=1}^{T}(h_t^+ + h_t^-)^2}}$$

where $h^+ = P_+ h$ and $h^- = P_- h$.

### D.3 HOUSEHOLDER PSEUDO-ROTATION

The steering function in activation editing maps an incoming hidden activation $h$ to a modified activation $\hat{h}$:

$$\hat{h} = f(h)$$

In the following, we instantiate this formulation for the Householder Pseudo-Rotation (HPR) method. HPR reflects and then rotates activations while preserving norm.

1. Reflection across a learned hyperplane.
2. Rotation on the 2D plane spanned by $(h, \dot{h})$.

The HPR steering function is:

$$f_{\mathrm{HPR}}(h) = \hat{\sigma}h + (1 - \hat{\sigma})\left[\frac{\sin(\gamma_1)}{\sin(\gamma_2)}\dot{h} + \frac{\sin(\gamma_2 - \gamma_1)}{\sin(\gamma_2)}h\right]$$

A linear probe is first trained:

$$f_{\mathrm{probe}}(h) = \sigma(\theta_{\mathrm{probe}}^\top h)$$

with loss:

$$\mathcal{L}_{probe} = \frac{1}{N}\sum_{i=1}^{N}\left[\mathrm{BCE}(\sigma(\theta_{\mathrm{probe}}^\top h_i^+), 1) + \mathrm{BCE}(\sigma(\theta_{\mathrm{probe}}^\top h_i^-), 0)\right]$$

Householder reflection uses:

$$H = I - \frac{2\theta_{\mathrm{probe}}\theta_{\mathrm{probe}}^{\top}}{\theta_{\mathrm{probe}}^{\top}\theta_{\mathrm{probe}}}$$

$$\dot{h} = Hh$$

A neural predictor gives rotation angle:

$$\gamma_1 = \pi \cdot \sigma(\mathrm{MLP}(h))$$

Angle supervision target:

$$g(h^+, h^-) = \arccos\left(\frac{(h^+)^{\top}h^-}{\|h^+\|\|h^-\|}\right)$$

Angle loss:

$$\mathcal{L}_{angle} = \frac{1}{N}\sum_{i=1}^{N}\left[(f_{\mathrm{angle}}(h_i^-) - g(h_i^+, h_i^-))^2 + f_{\mathrm{angle}}(h_i^+)^2\right]$$

Final rotation step:

$$\hat{h} = \frac{\sin(\gamma_1)}{\sin(\gamma_2)}\dot{h} + \frac{\sin(\gamma_2 - \gamma_1)}{\sin(\gamma_2)}h$$

Polarity decision:

$$\hat{\sigma} = \lfloor f_{\mathrm{probe}}(h)\rceil$$

# E  DISCUSSION

## E.1  SIGNIFICANCE OF SIMS

**Model steering** *versus* **self-critic RL**. Model steering learns lightweight functions that act on interme-diate activations of a *frozen* model at inference time. In contrast, self-critic RL optimizes a *trainable* policy via gradient updates on parameters using self-produced preferences or critiques. Practically, steering is plug-in and architecture-agnostic (no weight updates), whereas self-critic RL entails training dynamics (credit assignment, stability/regularization, exploration) and the compute/memory footprint of optimization.

SIMS *versus* **decoding-time value/constraint guidance**. Decoding-time guidance evaluates or scores tokens as they are generated and adjusts next-token probabilities at *every* step, coupling latency to sequence length and the cost of the auxiliary scorer/controller. SIMS instead *precomputes* layer-wise steering transforms that shift residual-stream activations during a forward pass. This yields an inference cost that scales with the number of layers via small matrix operations, without per-step scoring or backprop, and keeps memory stable across the decode.

**Steering functions** *versus* **LoRA**. LoRA adapts model *weights* via low-rank trainable matrices learned with backprop; deployment replaces or composes altered weights with the base model. Steering functions leave weights unchanged and operate on *activations* at run time. Consequently, LoRA requires task- or model-specific fine-tuning and checkpoint management, while steering can be learned once from activations and applied to frozen checkpoints with minimal integration overhead.

SIMS *versus* **label-free RL**. (1) *Learning paradigm.* Label-free RL learns or fine-tunes a policy using self-generated preference signals. SIMS learns small activation transforms that modulate internal representations at inference, leaving the underlying policy fixed. (2) *Compute/engineering cost.*

Label-free RL entails iterative gradient updates, rollouts, and stability controls; SIMS trains compact transforms (often linear/affine), then applies them as inexpensive forward operations. (3) *Objective focus.* Label-free RL optimizes a return defined by intrinsic or self-derived rewards, often seeking new capabilities or strategy shifts. SIMS targets *consistent behavioral alignment* by nudging hidden states toward desired manifolds and away from undesired ones, prioritizing controllability and efficiency over learning a new policy from scratch.

The fundamental trade-offs between model steering and alternative solutions are as follows. Model steering is lightweight, modular, inference-efficient, but may have limited expressiveness. LoRA/self-critic RL are more expressive but require parameter updates and higher computational costs.

This work represents a paradigm shift from existing model steering approaches. All prior methods (e.g., ActADD, CAA, HPR) require high-quality, externally annotated preference data (human-labeled positive/negative examples) that are often costly to obtain, error-prone, and directly constrain their applicability. SIMS eliminates this critical dependency by generating its own contrastive training signals through iterative self-evaluation. To realize this supervision-free framework, we develop several novel techniques: (1) self-generated contrastive sampling that creates training signals from the model's own outputs, (2) iterative refinement cycles that progressively improve steering effectiveness, and (3) prompt ranking and contrast sampling strategies that optimize sample quality without external guidance.

# F    USE OF LLM

We used a large language model (LLM) only for language editing (clarity, grammar, and tone). The LLM did not generate ideas, code, analyses, figures, tables, or experimental results. No proprietary or sensitive data were shared with the LLM. All mathematical statements, algorithmic descriptions, citations, and empirical results were written, verified, and are the responsibility of the authors. Model suggestions were reviewed by the authors for accuracy, and any references were independently checked. Further details are provided in the paper's supplementary materials.

